# Human Post-Translational SUMOylation Modification of SARS-CoV-2 Nucleocapsid Protein Enhances Its Interaction Affinity with Itself and Plays a Critical Role in Its Nuclear Translocation

**DOI:** 10.3390/v15071600

**Published:** 2023-07-21

**Authors:** Vipul Madahar, Runrui Dang, Quanqing Zhang, Chuchu Liu, Victor G. J. Rodgers, Jiayu Liao

**Affiliations:** 1Department of Bioengineering, College of Engineering, Bourns College of Engineering, University of California at Riverside, Riverside, CA 92521, USA; vmada001@ucr.edu (V.M.); rdang018@ucr.edu (R.D.); chuchu.liu@email.ucr.edu (C.L.); vrodgers@engr.ucr.edu (V.G.J.R.); 2Institute for Integrative Genome Biology, University of California at Riverside, Riverside, CA 92521, USA; quanqinz@ucr.edu; 3Department of Botany, College of Natural & Agricultural Sciences, University of California at Riverside, Riverside, CA 92521, USA; 4Biomedical Science, School of Medicine, University of California at Riverside, Riverside, CA 92521, USA

**Keywords:** SARS-CoV-2 N protein, SUMOylation, qFRET, protein interaction dissociation constant K_D_, nuclear translocation

## Abstract

Viruses, such as Severe acute respiratory syndrome coronavirus 2 (SARS-CoV-2), infect hosts and take advantage of host cellular machinery for genome replication and new virion production. Identifying and elucidating host pathways for viral infection is critical for understanding the development of the viral life cycle and novel therapeutics. The SARS-CoV-2 N protein is critical for viral RNA (vRNA) genome packaging in new virion formation. Using our quantitative Förster energy transfer/Mass spectrometry (qFRET/MS) coupled method and immunofluorescence imaging, we identified three SUMOylation sites of the SARS-CoV-2 N protein. We found that (1) Small Ubiquitin-like modifier (SUMO) modification in Nucleocapsid (N) protein interaction affinity increased, leading to enhanced oligomerization of the N protein; (2) one of the identified SUMOylation sites, K65, is critical for its nuclear translocation. These results suggest that the host human SUMOylation pathway may be critical for N protein functions in viral replication and pathology in vivo. Thus, blocking essential host pathways could provide a novel strategy for future anti-viral therapeutics development, such as for SARS-CoV-2 and other viruses.

## 1. Introduction 

Severe acute respiratory syndrome coronavirus 2 (SARS-CoV-2) caused a global pandemic responsible for the upper respiratory disease Coronavirus Disease 2019 (COVID-19). The rapid development of variants from the original strain with the onset of functional mutations highlights the need to discover its pathogenesis and a potential new strategy for anti-viral therapeutics development. The SARS-CoV-2 viral particle comprises the positive-sense single-strand RNA (~30 kb) genome, which encodes 29 proteins packed around nucleocapsid (N) proteins. The compacted RNA nucleocapsid complex is enveloped in a lipid membrane with embedded membrane proteins (M) and envelope proteins (E) [1]. The glycoprotein spike protein (S), exposed outside on the viral particle surface, is found to have a high affinity for human angiotensin-converting enzyme 2 (hACE2) [2].

The SARS-CoV N protein’s primary function is to package the ~30 kb single-stranded viral RNA genome (vRNA) into a ribonucleoprotein (RNP) assembly called the capsid, and interact with M and E proteins to help viral envelope formation and viral particle assembly [3,4,5]. Cryo-electron tomography (CryoET) studies have elucidated the complex structure of the N protein bound to vRNA and the N protein bound to the M protein. Recent investigations concluded that the oligomerized N protein compacts RNA into a structure, which forms a phase-separated condensate within the viral particle that binds to the M protein. A viral particle’s average diameter was found to be 80 nm and contained 30–35 vRNPs [1,6,7,8]. These investigators predicted that each vRNP is 15 nm in diameter, holds 12 copies of the N protein wrap, and is coated with viral RNA, creating a condensate that encapsulates the genomic vRNA by interacting with the M protein [6,9]. The sub-domains that enable protein–protein complexes of the N protein oligomers to form dimers and tetramers enable the complex formation of vRNP, which is ultimately packed within a virion.

The SARS-CoV N and SARS-CoV-2 N proteins have homology across their sequences and have been shown to have similar domains and functions [10]. The N protein-identified domains are the N-terminal domain (NTD from 1–50 aa), followed by the RNA binding domain (RBD from 51–174 aa), the dimerization domain (247–364 aa), and the C-terminal domain (CTD from 365–419 aa) [9]. The CTD was reported to interact with the M protein within the viral envelope and supports the vRNA super structure [6]. A linker region exists between the RBD and the dimerization domain (174–245 aa). This region is a serine- and arginine-rich region on the protein that is reported to be phosphorylated [11].

Investigations of N protein subcellular localization in the nucleolus and its interaction network with the host proteome have found that the functional attributes of the N protein are more than simply the vRNP complex. Although the coronavirus assembly is localized at the endoplasmic reticulum–Golgi intermediate compartment membranes, the SARS-CoV N protein is often found predominantly in the cytoplasm and the nucleus [12,13]. The SARS-CoV N protein is typically clustered in the nucleolus, a site for ribosome biogenesis, cell cycle regulation, and apoptosis [14]. The localization of the N protein in the nucleolus suggests that its role is to prevent host cell proliferation to favor virus RNA synthesis and assembly [14,15]. The anti-viral response interferon of host cells is inhibited by the N protein [16]. The N protein mediates the inhibition of interferon antiviral responses by sequestering activated STAT proteins within the cytosol and preventing IFN signaling from progressing [17].

The SARS-CoV N protein is reported to depend on post-translational modifications (PTMs) from the host proteome for its functional RNA binding properties. The PTM SUMOylation of the SARS-CoV N protein modulates its subcellular localization in cells [18]. A significant discovery that was attributed to the host PTM for viral infection is the increase in SARS-CoV N protein dimerization after the overexpression of the SUMO1 gene [18]. It has been demonstrated that N proteins can form superstructures of dimers and tetramers, and ultimately oligomers, but the mechanism underlying oligomerization has yet to be determined. The oligomerization of the N protein is a critical factor in viral genome packaging [6,8]. A recent PTM study on the SARS-CoV-2 N protein has found potential phosphorylation of serine 197 and threonine 205 [11]. The study observed SARS-CoV-2 N protein modulation of RNA binding with mutations at 197 and 205 in the S/R-rich region. The phosphorylation modification is inherent to the protein’s function and infers the virus progression’s dependence on host PTMs [6,7]. Furthermore, an investigation of N protein ubiquitination in cells observed lysine 169, 374, and 388 to be ubiquitin-modified. However, there was no follow-up functional study [19].

Förster resonance energy transfer (FRET) has been widely used in biological and biomedical research to detect spatial and temporal molecular interactions in vitro and in vivo [20,21]. FRET assays are highly sensitive and can detect molecules within 1–10 nm. Several attempts have been made to develop the FRET assay into a quantitative measurement of protein–protein interaction affinity [22,23,24]. However, progress has been slow due to complicated procedures or challenges in differentiating the FRET signal from other direct emission signals from the donor and receptor. We recently developed a quantitative FRET (qFRET) analysis based on cross-wavelength correlation coefficiency and its application in determining protein–protein interaction affinity (K_D_). This was carried out with FRET acceptor emission and donor quenching methods [25,26,27]. This method has been successful in determining protein interaction affinity and reaction intermediates for several systems [28].

The SUMO modification of target proteins increases the target protein’s affinity to other cellular proteins and changes the protein subcellular localization [29,30,31,32]. In this study, we use an in vitro qFRET assay for the SUMOylation of the SARS-CoV-2 N protein coupled with mass spectrometry to identify the sites of SUMO-modified lysine. When mutated to Arg, one of the modified lysine sites, Lys65, restricted the N protein in the cytosol, suggesting that SUMOylation may play a critical role in the nuclear translocation of the N protein. In addition, we found SUMOylated N proteins have a much higher affinity to interact with each other than un-SUMOylated N proteins, suggesting a gain-of-activity of SARS-CoV-2 N protein from host PTM. These results suggest that host SUMOylation may play essential roles in the SARS-CoV-2 life cycle and could be a potential target for future therapeutic development.

## 2. Materials and Methods

### 2.1. Expression and Purification of SUMOylation Enzymes and SARS-CoV-2 N Protein

The in vitro qFRET SUMOylation reaction was completed with the E1, E2, and E3 enzymes in the SUMOylation cascade. The E1 activation enzyme complex, UBA2 and AOS1, E2 conjugating enzyme UBC9, and E3 ligase PIAS1 were all cloned into the pET28B vector for expression in BL_21_(DE_3_) cells. The FRET pairs CyPet and YPet were N-terminal tagged to SUMO1 and the substrates, respectively, and cloned into pET28B for expression in BL_21_(DE_3_). The BL_21_(DE_3_) cell line with each gene was inoculated at 1:100, grown at 37 °C to OD of 0.4 at 600 nm, and then induced overnight at 22 °C with 0.25 mM IPTG. The cells were lysed (Lysis buffer contains 20 mM Tris-HCl (pH 7.5), 0.5 M NaCl, 5 mM Imidazole) by sonication and centrifuged at 35,000× *g*. The soluble fraction was purified by 6×His tag to Ni-NTA beads affinity chromatography through a gravity column. The bound proteins were washed with Buffer 1 (20 mM Tris-HCl (pH 7.5), 0.3 M NaCl), Buffer 2 (20 mM Tris-HCl (pH 7.5), 1.5 M NaCl, and 0.5% Triton X-100), and Buffer 3 (20 mM Tris-HCl pH 7.5, 0.5 M NaCl, and 10 mM Imidazole). The proteins were eluted using the following Elution buffer (20 mM Tris-HCl, 300 mM NaCl, and 450 mM Imidazole) and dialyzed in 20 mM Tris-HCl (pH 7.5), 50 mM NaCl, and 1 mM DTT.

### 2.2. In Vitro SUMOylation Assay

The in vitro SUMOylation assay is completed with 6xHisCyPet-SUMO1 500 nM, 6×HisYPet-N protein wildtype 2000 nM, E1 hetero-dimer AOS1/UBA2 at 100 nM, E2 conjugating enzyme UBC9 200 nM, E3 ligase PIAS1 250 nM, and in SUMOylation buffer (20 mM Tris-HCl (pH 7.5) 50 mM NaCl, 4 mM MgCl_2_, 1 mM DTT). Functional controls are implemented for observing non-specific interaction by a negative control reaction without 2 mM adenosine triphosphate (ATP). Each reaction was incubated at 37 °C for 60 min and measured in a 384-well microplate (Grenier, Monroe, NC, USA). The FRET wavelengths, E_mTotal_, are 414 nm excitation and 530 nm emission, Fl_DD_, 414 nm excitation and 475 nm emission, and FL_AA_, 475 nm excitation and 530 nm emission. The quantitative E_mFRET_ parameters, α of 0.34 +/− 0.003, and β of 0.003 +/− 0.001 variables, are determined using the formulation outlined in previous work from Yang et al. [25]. Equation (1) provides the calculation of E_mFRET_ that quantifies the FRET signal by subtracting free donor and acceptor emissions from the total fluorescence emission.
E_mFRET_ = (E_mTotal_) − (α(FL_DD_) + β(FL_AA_))(1)
where FL_DD_ is fluorescence emission of the donor when excited at the donor excitation wavelength, and FL_AA_ is fluorescence emission of the acceptor when excited at the acceptor excitation wavelength.

The specificity of SUMO protein to the SUMOylation target can potentially yield a false positive FRET response. Thus, parallel functional controls of reactions without ATP were implemented to observe differences between ATP and no ATP. Samples from each plus or minusATP/plus or minusE3 reaction were also immunoblotted with anti-SUMO1 monoclonal antibody (Santa Cruze Biotechnology, Santa Cruze, CA, USA).

### 2.3. Mass Spectrometry Analysis to Determine SUMO-Modified Lysine on N Protein

For the in vitro SUMOylation reactions, the substrate, YPet-SARS-CoV-2 Nucleocapsid protein, was added at 3000 nM, and CyPet tagged SUMO1 protein were added at 1000 nM. Activating Enzyme Complex 1 (E1) is at 100 nM, and Conjugating Enzyme 2 (E2) at 100 nM, E3 ligase at 500 nM in SUMOylation buffer (20 mM Tris-HCl (pH 7.5) 50 mM NaCl, 4 mM MgCl_2_, 1 mM DTT) and 2 mM ATP. The reactions took place at 37 °C for 4 h. The in-solution proteolytic digestions were performed with Pierce^TM^ Glu-C Protease. Samples were digested at a 1:100 ratio for substrate-to-enzyme ratio and ran overnight (16 h) at 37 °C. Each completed digestion was acidified to a final concentration of 0.1% *v*/*v* TFA, speed vacuumed to dry the product, and then reconstituted to 0.1% *v*/*v* TFA for Mass spectrometry (MS) loading.

### 2.4. LTQ Orbitrap Xl Loading and Run

Samples consisted of approximately 1 μM of in-solution digested product. For each proteolytic, enzyme digestion liquid chromatography was performed on a Thermo nLC1200 (Thermo Fisher Scientific Inc., Waltham, MA, USA) in single-pump trapping mode with a Thermo PepMap RSLC C18 EASY-spray column (2 μm, 100 Å, 75 μm × 25 cm) and a Pepmap C18 trap column (3 μm, 100 Å, 75 μm × 20 mm). The solvents used were A: water with 0.1% formic acid and B: 80% acetonitrile with 0.1% formic acid. Samples were separated at 300 nL/min with a 250 min gradient starting at 3% B increasing to 30% B from 1 to 231 min, then to 85% B at 241 min, holding for 10 min.

Mass spectrometry data were acquired on a Thermo Orbitrap Fusion mass spectrometer (Thermo Fisher Scientific Inc., Waltham, MA, USA) in data-dependent mode. A full scan was conducted using 60 k resolution in the Orbitrap in positive mode. Precursors for MS^2^ were filtered by monoisotopic peak determination for peptides, intensity threshold 5.0 × 10^3^, charge state 2–7, and 60 s dynamic exclusion after 1 analysis with a mass tolerance of 10 ppm. Higher-energy C-trap dissociation (HCD) spectra were collected in ion trap MS^2^ at 35% energy and isolation window 1.6 *m*/*z*.

### 2.5. Bioinformatic Analysis of MS Data

The LTQ-orbitrap XL raw data were analyzed on Thermofisher Proteome Analyzer^TM^ (Thermo Fisher Scientific Inc., Waltham, MA, USA). Each protein’s complete amino acid sequence was provided as a reference for analysis. The expected SUMOylated lysine proteolytic products were searched for using both software suites, matched to mass over charge spectrums within the Thermofisher and tabulated (Table 1) Proteome Analyzer^TM^. Precursor ion peptide tolerances were set at 5 ppm, and MS/MS peptide tolerances were set at 1 Dalton.

### 2.6. Validation of SUMOylation Sites of SARS-CoV-2 N Protein Using Engineered SUMO1 Peptide

To accurately determine the SUMOylation sites using mass spectrometry, we engineered the SUMO1 peptide by changing the T95R for trypsin cleavage to generate smaller peptides for detection. The cDNA of SUMO1T95R mutant was generated by PCR with the forward primer (5′-gtcgacatgtctgaccaggaggcaaaacctt-3′) containing the SalI restriction site and the reverse primer (5′-gcggccgcctaaccccccctttgttcctgata-3′) containing the NotI restriction site and stop codon. The SUMO T95R mutant construction was generated by inserting cDNA in pET28b with CyPet labeled in SalI and NotI. The protein expression was conducted according to the same procedure as above.

After the SDS-PAGE gel of the SUMOylation sample, the gel was cut into 1 mm^3^ cubes. Then, the cubes were destained sequentially with 500 μL 25% and 50% acetonitrile (ACN) in 50 mM ammonium bicarbonate (pH 7.8). Each cube was sonicated twice for 15 min and the liquid was discarded. A 0.5–1 μL 100% ACN was added, followed by sonication for 10 min and dried using a Speed-Vac. A 200 μL of the 20 mM DTT in 50 mM ammonium bicarbonate was added and incubated at 37 °C for 1 h. Then, 49 μL of the 500 mM iodoacetamide was added to the sample and incubated for 30 min in the dark at room temperature. The gel particles were washed with 500 μL 100% ACN twice and dried with a Speed-Vac. The proteins were then digested in gel with trypsin for 16 h at 37 °C with an enzyme/protein ratio of 1:100. Peptides were extracted from the gel pieces with 25% ACN/5% acetic acid (HOAc). The extracted peptides were concentrated using a Speed-Vac. The peptides were desalted by using C18 ZipTip. The peptide solution was then dried by Speed-Vac and stored at −80 °C until analyzed by LC-MS/MS.

The peptides were resuspended with 20 μL water with 0.1% formic acid, separated by nano-LC, and analyzed by online electrospray tandem mass spectrometry. The experiments were performed on an EASY-nLC 1200 system (Thermo Scientific Inc., Waltham, MA, USA) connected to a quadrupole-Orbitrap Fusion Tribrid Mass Spectrometry (Thermo Fisher Scientific Inc., Waltham, MA, USA) equipped with an EASY-Spray ion source. Five μL peptide sample was loaded onto the trap column (Thermo Scientific Acclaim PepMap C18, 75 μm × 2 cm) with a flow of 10 μL/min for 3 min and subsequently separated on the analytical column (Acclaim PepMap C18, 75 μm × 25 cm) with a linear gradient, from 3% D to 40% D in 55 min. The column was re-equilibrated to initial conditions for 5 min. The flow rate was maintained at 300 nl/min, and the column temperature was maintained at 45 °C. An electrospray voltage of 2.2 kV above the inlet of the mass spectrometer was used.

The Orbitrap Fusion mass spectrometry was operated in the data-dependent mode to switch automatically between MS and MS/MS acquisition. Survey full-scan MS spectra (*m*/*z* 375–1500) were acquired with a mass resolution of 60 K, followed by fifteen sequential high energy collisional dissociation (HCD). The AGC target was set to 4 × 10^5^, and the maximum injection time was 100 ms. The MS/MS acquisition was performed in the ion trap. The AGC target was set to 3 × 10^4^, and the isolation window was 1.6 *m*/*z*. Ions with charge states 2+, 3+, and 4+ were sequentially fragmented by higher energy collisional dissociation (HCD) with a normalized collision energy (NCE) of 35%, and the fixed first mass was set at 100. One micro scan was recorded using the dynamic exclusion of 30 s in all cases.

The two raw data sets of human samples were processed and analyzed using MaxQuant (version 2.1.4.0, Max Planck Institute of Biochemistry, Martinsried, Germany), with the N/M1 protein database. In particular, mass tolerances for precursor and fragment ions were 6 and 10 ppm, respectively. The minimum peptide length was 6 amino acids, and the maximum number of missed cleavages for trypsin was 2. Carbamidomethyl (C) was set as fixed modifications. Oxidation (M) and acetyl (Protein N-term) were set as variable modifications.

### 2.7. Construction and Design of N Protein Lysine to Arginine Mutants

The mass spectrometry results provided a total of four SUMOylated lysine residues. Among 31 lysine residues on the SARS-CoV-2 N protein, lysine 61, 65, and 347 were found to be SUMOylated and received reconfirmation in the in vitro reaction. The mutant DNA templates were constructed through PCR, with point mutations at the lysine to arginine coding sequences. The primers for the mutations are listed in Table 2. The final Gibson reaction of bacterial expression pET28B vector SalI and NotI was created for mammalian expression pCDNA3.1-FLAGtag-Nprotein-YPet. Tabulated PCR primers are shown in Table 3 for pcDNA3.1.

### 2.8. In Vitro SUMOylation with qFRET Reporter for N Protein Mutants

The in vitro SUMOylation assay of SARS-CoV-2 N protein mutants was an initial screening to determine the impact of lysine sites on SUMOylation. The assay was set up at the same concentration as the optimized conditions, 6×HisCyPet-SUMO1 500 nM, 6×HisYPet-N Protein wildtype and mutants 2000 nM, E1 hetero-dimer AOS1/UBA2 at 100 nM, E2 conjugating enzyme UBC9 200 nM, E3 ligase PIAS1 250 nM, and SUMOylation buffer of 20 mM Tris-HCl (pH 7.5) 50 mM NaCl, 4 mM MgCl, and 1 mM DTT. Functional controls were put in place for non-specific interaction, as a negative control reaction without ATP, and to observe a significant boost in FRET, a control reaction without E3 ligase. Each reaction was incubated at 37 °C for 60 min. Using Equation (1), we determined E_mFRET_ from the three measured fluorescence emissions, E_mTotal_, FL_DD_, and FL_AA_. The measurements were taken on Molecular Devices Spectra M3^TM^, with “Endpoint” settings, and PMT at constant gain set to “Low”.

### 2.9. K_D_ Determination of SUMOylated or Not-SUMOylated N Proteins Using qFRET

The evaluation of N protein oligomerization by in vitro qFRET-based K_D_ affinity assay. The individual N protein wild type and four Lys mutants (K to R) at 61, 65, 347, and 355 were first cloned into the FRET donor and acceptor genes, pET28(b)-CyPet or pET28(b)-CyPet, respectively, as fusion proteins for the FRET assay. Each N protein was tagged with the donor or acceptor pair fluorescent proteins, CyPet and YPet, respectively, to implement the qFRET assay. After expression in Bl21(DE3) and purification using Ni+ beads, each pair of proteins was SUMOylated in an in vitro SUMOylation assay containing SUMO enzymes and SUMO1 as follows; 6×His-SUMO1 6 μM, E1 hetero-dimer 6×His-AOS1/UBA2 at 100 nM, 6×His-E2 conjugating enzyme UBC9 200 nM, E3 ligase 6×His-PIAS1 250 nM as well as 6×HisYPet-N or 6×His-CyPet proteins at 6 μM in a SUMOylation buffer (20 mM Tris-HCl (pH 7.5) 50 mM NaCl, 4 mM MgCl_2_, 1 mM DTT). Negative control, without ATP, was set up for non-SUMOylated proteins. The SUMOylation reaction was incubated at 37 °C for 1 h.

The SUMOylated N proteins were directly used to determine interaction affinities [25,26]. The qFRET-based K_D_ assay contained the donor fusion protein concentration at 0.5 μM, and the acceptor was titrated from 0 μM to 2.5 μM. The series of titrations were individually performed to determine the E_mFRET_ values of each YPet-N protein using Equation (1) and the three wavelengths, E_mTotal_, FL_DD_, and FL_AA,_ for each reaction. Again, the measurements were determined on a Molecular Devices Spectra M3^TM^, with “Endpoint” settings and PMT at constant gain set to “Low”. The K_D_ was then determined by fitting the data against Equation (2) using a nonlinear multi-regression method using Prism5^TM^(Graphpad Software, Boston, MA, USA),. Equation (2) is
(2)$$E_{\mathrm{mFRET}}=E_{{\mathrm{mFRET}}_{\mathrm{Max}}}\times \left(\frac{{\left[\mathrm{Acceptor}\right]}_{\mathrm{Total}}-{\left[\mathrm{Donor}\right]}_{\mathrm{Total}}-K_{D}+\sqrt{{\left({\left[\mathrm{Donor}\right]}_{\mathrm{Total}}+K_{D}-{\left[\mathrm{Acceptor}\right]}_{\mathrm{Total}}\right)}^{2}+4\times K_{D}\times {\left[\mathrm{Acceptor}\right]}_{\mathrm{Total}}}}{{\left[\mathrm{Donor}\right]}_{\mathrm{Total}}+K_{D}-{\left[\mathrm{Acceptor}\right]}_{\mathrm{Total}}+\sqrt{{\left({\left[\mathrm{Donor}\right]}_{\mathrm{Total}}-{\left[\mathrm{Acceptor}\right]}_{\mathrm{Total}}+K_{D}\right)}^{2}+4\times K_{D}\times {\left[\mathrm{Acceptor}\right]}_{\mathrm{Total}}}}\right).$$

The constraints for the non-linear regression fit were set to have a donor concentration at a constant of 500 nM, and the values for K_D_ and E_mFRET_Max__ were constrained to be nonzero. 

### 2.10. SARS-CoV-2 N Protein Aggregation Assay with or without SUMO Modification

All components of the SUMOylation assay (2.5 µM CyPet-SUMO1, 1 µM E1 (Aos1 and Uba2), 1.5 µM E2 Ubc9, 1.5 µM E3 PIAS1 and 1 µM SARS-CoV-2 N protein) were combined in SUMOylation buffer containing 50 mM Tris-HCl pH 7.4, 1 mM DTT, and 4 mM MgCl_2_ in a total volume of 200 µL. A concentration of 2 mM ATP was added to each sample and the mixtures were incubated in an Eppendorf tube at 37 °C for 90 min. Two groups of SUMOylation samples (one has 2 mM ATP, another one without ATP) were added 50 µL of CHAPS buffer (20 mM HEPES-KOH pH 7.5, 5 mM MgCl_2_, 0.5 mM EDTA, 0.1 mM PMSF and 0.1% CHAPS) containing 4 mM of disuccinimidyl suberate (dissolved in DMSO). Incubation took place for 40 min at room temperature to cross-link proteins.

For immunoblot assays of cross-linked N proteins and 250µL of cross-linked SUMOylation, samples were mixed with 50 µL 6×protein loading buffer and heated for 5 min at 90 °C. Then, 30 µL of protein samples were loaded on the 7.5% SDS-PAGE. The samples were separated using a constant 70 V until the protein loading buffer reached the end of the gel. The samples were then transferred at 25 V for 4 h using a wet transfer system. The membrane was blocked in blocking buffer (1× TBST, 5% BSA, 0.02% sodium azide) for 1 h at 4 °C, then incubated 1 h at room temperature with room temperature anti-SARS-CoV-2 N antibody (R&D systems, Clone 1035111, MAB10474-SP, 1:400) diluted in blocking buffer. The membrane was incubated 1 h at room temperature with anti-mouse IgG, HRP (CST, 7076) in blocking after TBST washing 3 times. Images of the results were obtained using the BioSpectrum^®^ Imaging System (UVP, Upland, CA, USA).

### 2.11. Cellular Translocation of N Protein

Immunostaining of N protein was used to investigate the dependency of SUMOylation of N protein on translocation between cytosol and nucleus. Glass coverslips were coated with L-lysine overnight at 22 °C under UV light in a 12-well plate. Post-coating HUH7 cells were seeded onto the coverslips and grown until 50% confluent. The cells were transfected with pcDNAD3-Nwt, -N K61R, -K65R, and -K347R plasmids, respectively. After 24 h of transfection, the cells were washed with DPBS and fixed in 4% paraformaldehyde (PFA) for 15 min with rocking. After fixing, the PFA was aspirated, and the cells were washed with DPBS. The cell nucleus was stained with Hoechst 33342 for incubation of 15 min. After nuclear stain, the cells were washed 4 times with DPBS for 5 min incubation. The cells were imaged on Olympus BX43 (Evident Scientific, Waltham, MA, USA), and images were stacked and analyzed using ImageJ software (Version 1.54f, https://imagej.net/ij/download.html) (accessed on 20 May 2021).

### 2.12. Statistical Analysis

The post-analysis was completed on GraphpadPrism7^TM^, using a one-way ANOVA with post hoc Tukey test. Samples minus ATP were used as the control group.

The student *t*-test was performed using GraphPad web server https://www.graphpad.com/quickcalcs/ttest2/ (accessed on 20 May 2021).

## 3. Results

### 3.1. In Vitro qFRET Assay for SUMOylation of SARS-CoV-2 N Protein

We developed an in vitro qFRET-based SUMOylation assay that includes the SUMO E1 activating enzyme, E2 conjugation enzyme, and E3 ligase. In this assay, the SUMO peptide was fused with the FRET donor, CyPet, and the substrate SARS-CoV-2 was fused with the FRET acceptor, YPet (Figure 1A). In the presence of SUMOylation enzymes, E1, E2, and E3, the CyPet-SUMO1 is conjugated to the YPet-N protein, leading to a FRET signal (Figure 1B). We validated this assay using a classical Western-blot assay using an anti-SUMO1 antibody (Figure 1C). The SUMOylation of N protein was catalyzed by the E1 and E2, but significantly enhanced by E3 ligase, PIAS1, indicated by the disappearance of free CyPet-SUMO1(Figure 1C). The results suggest that the FRET-based SUMOylation assay is feasible and efficient.

### 3.2. Mass Spectrometry Analysis to Determine SUMO Modified Lysine on SARS-CoV-2 N Protein

We then conducted the SUMOylation assay of the SARS-CoV-2 protein using qFRET to follow the SUMO peptide conjugation. The observed E_mFRET_ signal for the SUMOylation of SARS-CoV-2 N protein showed some difference but not a significant increase in E_mFRET_ without E3, but a significant increase in the presence of E3 (Figure 2A). A *t*-test analysis was completed on the triplicate measurements, compared with and without ATP and with and without E3 ligase. There was also a significant E_mFRET_ signal increase in the presence of E3 ligase, suggesting a robust SUMOylation event mediated by the E3. A one-way ANOVA was used to analyze the significance between the control group without ATP (-ATP), with ATP and no E3 ligase, and with ATP and E3 ligase.

We then determined the SUMOylation sites of the SARS-CoV-2 N protein using mass spectrometry (Thermo Orbitrap Fusion). The identified modified lysine residues are illustrated in Figure 2B, along with the other 27 lysines found unmodified by the SUMO1. The first mass spectrometry experiment covered 95% of the N protein. The spectrum of the identified peptide with Lys61 modification was a large section of N protein7, which had Lys61 and 65 within it. To validate the SUMOylation sites of SARS-CoV-2 N protein, we conducted a series of mass spectrometry studies to validate the SUMOylation sites using an engineered SUMO1 peptide, T95R, as the trypsin would cleave at the C-terminus of the R and leave a short peptide GG, making the identification of SUMOylation sites more robust. A Lys residue identified as SUMO1 modification is Lys61, and the identified peptide from 41–61 matches the expected trypsin cut (Figure 2C). However, Figure 2D has a similar peptide with lysine-65 holding the SUMO1 modification. The trypsin-cleaved peptide was like the previous peptide, N protein position −41–65. The N protein peptide from positions 339–355 showed SUMO1 modification GG residue (Figure 2E). This peptide matched the trypsin cut pattern at positions 339–355.

The secondary analysis of each lysine site can be performed by evaluating each site matching the SUMO consensus motif. Based on the SUMO consensus motif of a hydrophobic residue (Ψ), the modified lysine (K), any amino acid, and either an aspartic acid or a glutamic acid. Numerous groups have applied the Ψ-K-x-D/E motif to ascertain the SUMOylation site. Only two servers, GPS-SUMO and JASSA, were used to determine if these sites matched the SUMO consensus [33,34]. Both servers only pointed to lysine position 338 as the highest probability, and lysine 61, 65, and 347 all were low probability. This result did not match the SARS-CoV-1 N protein SUMOylation site and was therefore not included in this study.

### 3.3. qFRET Assay for In Vitro SUMOylation Assay of N Protein Mutants

The results of the in vitro SUMOylation of N protein with qFRET as a reporter resulted from the conjugation of CyPet-SUMO1 to YPet-N was then performed to show the SUMO1 conjugation (Figure 3A). We observed significant signal from wildtype N protein in the presence of E3 ligase PIAS1 and ATP, indicating a significant activity of PIAS1 ligase for N protein SUMOylation (Figure 3A). In comparison with the wildtype N protein, the K61R and K65R mutants showed a significant drop in the signal without E3. This was nearly equivalent to the control without ATP, which showed a small signal reduction with E3. This pattern was the same as observed in the double K61 and K65 mutant without E3 ligase. However, both K61R and K65R still showed significant FRET signals with PIAS1, indicating SUMOylation occurred at other Lys sites in which PIAS1 might mediate. The K347R mutant of the N protein also showed significant FRET signals, suggesting other Lys sites, such as K61 and K65, were still SUMOylated. The one-way ANOVA analysis with Tukey test showed significant differences when E3 was added across all reactions.

We also confirmed the SUMOylation results from the qFRET assay using the Western-blot analysis for the N protein SUMOylation (Figure 3B) and its mutants (Figure 3C). The shifted band corresponding to the SUMOylated N protein only occurred in the presence of PIAS1 and ATP in the Western-blot using anti-N antibody. This is inconsistent to the results from the qFRET assay (Figure 3A). In the mutant protein Western-blot assay, no single Lys mutation interrupted the SUMOylation of the N protein, consistent with the results from the qFRET assay.

### 3.4. K_D_ Determination of SUMOylated or Not SUMOylated N Protein Using qFRET

The SARS-CoV-2 N proteins interact with each other to form oligomers for viral genome RNA packaging. Thus, protein–protein interaction of the N protein is critical for its functions in vivo. We, therefore, performed qFRET assays to determine the interaction affinities of N wildtype and Lys mutant proteins with or without SUMOylation. One of the advantages of our qFRET technology is that it does not require pure proteins for K_D_ determination. Therefore, we first conducted the in vitro SUMOyaltion of all N proteins with CyPet-SUMO1 and YPet-N proteins and followed with the K_D_ determinations without purification.

The E_mFRET_ response from the titration of total acceptor protein and the regressed fit was plotted in Figure 4. The E_mFRET_ of wildtype N protein from the titration of acceptor fusion protein YPet-WT N was plotted in points (circle/orange) for unmodified and (diamond/green) for SUMO1 modified N proteins (Figure 4A). We determined a K_D_ value of wildtype N protein self-interaction after SUMO1 modification to be 0.34 µM and standard error of 0.07, and 0.82 µM standard error of 0.18 when not SUMO modified (Figure 4B). A *t*-test for the K_D_ suggested their difference was statistically significance (*p* value of 0.016). This result suggests that SUMO modification of SARS-CoV-2 N protein increases its interaction affinity with itself, and this modification may facilitate the oligomerization of N protein in the nucleus for viral RNA genome packaging.

To validate the idea of SUMOylated SARS-CoV-2 N protein increasing its interaction affinity with itself, we conducted an aggregation experiment of the N proteins with or without SUMOylation. The N proteins were SUMOylated in the presence of SUMO1, Aos1/Uba2, Ubc9, and PIAS1 were SUMOylated in the presence or absence of ATP. The N proteins were crosslinked using disuccinimidyl suberate and immunoblot using an anti-N protein antibody. The result shows that SUMOylated N protein had more aggregates than the un-SUMOylated N protein (Figure 4C). This result supports the above interpretation that SUMOylation of the N protein increases its affinity for oligomerization.

We then tried to determine that each Lys residue involved in SUMOylation contributes to the increase in interaction affinity. The N protein mutants at each Lys residue position, 61, 65, and 347, were mutated to Arg, then expressed and followed by SUMOylation. The corresponding K_D_ values for both modified and unmodified N mutant proteins, with or without SUMOylation modification, were determined. The results of the qFRET K_D_ assay on the N protein mutants are listed in Table 1. The E_mFRET_ values spanned between 0 to 2.5 µM for YPet-N protein with and without SUMOylation. The affinity of the un-modified N proteins ranged from 0.92 to 1.98 µM, and the range for modified N proteins was 0.64 to 1.84 µM for the three Lys residues (Figure 4D,E). The results show a general increase in the interaction affinity with SUMOylation modification.

### 3.5. Nucleus Translocation of N Protein

We then examined the potential role of SUMOylation on each Lys site for its nucleus translocation of the SARS-CoV-2 N protein using a YPet-N fusion protein method. HUH7 cells were transfected with pcDNA3-N WT or individual 3 Lys mutant N protein plasmids. The YPet is an imaging marker of fusion protein YPet-N to track the sub-cellular localization of wildtype or Lys mutant N protein. The representative images from each slide are shown in Figure 5. The top row was the nuclear stains using Hoechst that provided the location of the nucleus (Figure 5 Top row). The images in the second row were obtained through the YPet channel, where we observed the YPet fused N protein and the third row was overlapped images of Hoechst staining and the YPet image. The YPet-N fluorescence image showed universal distributions among the whole cells, including cytosol and nucleus, but with very obvious condensed granules (Figure 5, 1st column). These fluorescent granules forming within the cell are inherent to the N protein function as oligomers. The N protein wildtype, or K61R or K347 mutants, were all observed as granules in both the cytosol and nucleus (Figure 5, 3rd row). Interestingly, N K65R protein was predominant in the cytosol, suggesting that Lys residue 65 as a SUMOylation site may be essential for its nucleus localization (Figure 5, 3rd row). This result indicates that SUMOylation of N protein may play a critical role in its nucleus translocation, viral RNA genome packaging, and subsequent viral particle assembly.

## 4. Discussion

Viruses take advantage of host factors for their infection, replication, virion assembly, and budding for their amplification and, consequently, pathogenesis. Thus, targeting host factors as a new strategy for anti-virus therapeutics is very promising [35,36,37,38,39]. Various viruses have extensively utilized human SUMOylation, including influenza A/B virus, HIV, and Ebola viruses [40,41,42]. We identified the SUMOylation sites of SARS-CoV-2 N protein, which is critical for viral genome RNA packing and viral–host interactions, using an in vitro qFRET-MS coupled approach. The N protein is found to crosslink at high concentrations in the cell but forms oligomers and dimers [3,4,43]. The overall organization of the vRNP complex formation by the N protein and viral RNA requires non-covalent interaction of the various forms of N protein. We demonstrated here that the affinity increases and nucleolus/nucleus localization of SARS-CoV-2 N protein with the SUMO modification may contribute significantly to its oligomerization and pathogenesis.

We identified and confirmed three SUMOylation sites, K61, 65, and 347, of SARS-CoV-2 N protein using the in vitro SUMOylation assay in the presence of SUMO E1, E2, and E3 enzymes. The qFRET-based SUMOylation assay, used alternatively was more sensitively and robustly than immunoblot, and is also used in evaluating covalent modifications of proteins. We found that the affinity of SARS-CoV-2 N protein with itself increased significantly from 0.82 ± 0.18 μM to 0.34 ± 0.07 μM. The SARS-CoV N protein was shown to form dimers and oligomers through its C-terminal 285–422 residues, including Lys 347 [44,45]. There may be other Lys sites in the region in which SUMOylation may also contribute to dimerization.

Previous studies indicated that the SARS-CoV N protein was predominately localized in the cytoplasm and only overexpressed N proteins localized in both the cytoplasm and nucleus [12,13,46]. Our bioinformatics study suggested that the SARS-CoV-2 N protein contains classical nuclear localization signals (NLS) as monopartite and bipartite motifs, implying that the N protein has the physicochemical property to translocate in the nucleus [13,46]. Our subcellular localization study of fluorescence protein-tagged N protein forms bright spots as granules in both cytoplasm and nucleus (Figure 5). We observe the yellow fluorescence protein-tagged wildtype N protein localized in both the cytosol and the nucleus, consistent with the previous report of SARS N protein [13]. The nuclear translocation of the SARS-CoV-2 N protein was abolished with Lys 65 mutation, suggesting a critical role of SUMOylation in protein nuclear translocation. A previous study using various truncated N proteins fused with GFP for a subcellular localization study showed both NLS 2 (aa 257–265) and NLS 3 (aa 369–390) localized to the cytoplasm and nucleolus [14]. SUMOyaltion has been shown to regulate cytoplasm/nuclear translocation either through the promotion of nuclear import or inhibition of nuclear export through nuclear localization signals (NLS), recognized by importins, and nuclear export signals (NES), recognized by exportins, respectively [31,32]. For example, the Ran-GTPase-activating protein RANGap1 and the MAP kinase ERK5 depend on SUMOyaltion for the nuclear translocation through their NLS. The transcription factor NFAT1 in T cell activation, once imported into the nucleus through dephosphorylation by calcineurin, is retained in the nucleus requires SUMOylation close to its NES [47,48,49,50]. Because there is no NES in the SARS-CoV-2 N protein, SUMOyaltion-dependent nucleus translocation of SARS-CoV-2 N protein may translocate through its NLS.

The SARS-CoV-2 proteome is relatively new to the scientific community, and the evaluation of protein post-translational modifications can help their impact on protein properties and viral pathogenesis. Demonstrated here is a versatile method to evaluate the covalent modification and non-covalent interactions using the same platform. An advantage of the in vitro evaluation of SUMOylation modifications is that it also provides clues for in vivo validations. Due to the numerous in-cell lysine modifications, the probability of missing or false negative classification of a lysine modification is high. Furthermore, the yield of a modified protein from an in-cell pull-down assay can be challenging and have low yields. Thus, researchers look to overexpress the SUMOylation within a cellular environment. However, overexpressed SUMOylation can bring about unwanted consequences to the cellular proteome. The in vitro qFRET assay used here provided a fluorescent reporter for SUMOylation and was directly used to identify modified lysine residues. The coverage of the protein identified in MS was up to 95%, and the other 5% is assumed to be degraded during the digestion and sample preparation. The high certainty and overall protein coverage in MS analysis provide confidence in the identified SUMOylation sites.

The reconstitution of SUMOylation reaction with qFRET as a reporter is a robust and rapid method for identifying SUMOylation events, characterizing enzymatic activity, and identifying SUMOylation sites [25,51,52]. This method includes the E3 ligase PIAS1 for its enhanced SUMOylation activity and is coupled with mass spectrometry to provide insight into identifying multiple SUMOylation sites [53,54,55]. The qFRET assay utilizes a FRET-optimized donor fluorescent protein tag, CyPet, on the SUMO1 protein and FRET optimized acceptor fluorescent protein tag, YPet, on the N-protein [56]. The CyPet-YPet fluorescent proteins experience the non-radiative FRET phenomenon when within a 10 nm distance between them, which is applicable for observing SUMO1 attachment. The FRET phenomenon has been described and applied in various protein–protein interaction studies, especially to measure molecule distance. FRET efficiency is proportional to the distance of two fluorophores (r) and the signal decays at a distant dependent rate of r^6^ between the two fluorophores [57,58,59]. We have developed a fluorescence-based method for determining the covalent attachment of Cypet-SUMO1 to the YPet-N protein. The method demonstrated here applies a “three cube FRET” fluorescence reporter that extracts the emission of the FRET signal, E_mFRET_, from the raw fluorescent signal, E_mTotal_, at the FRET wavelength. The extraction applies three different fluorescent measurements (Equation (1)) to extract the E_mFRET_ response from a FRET reaction. The method filters out cross channel signal of the unbound donor or acceptor from the FRET wavelength. The relationship found in Equation (1) determines the contribution of cross-talk from both the acceptor and donor by applying ratiometric constants alpha (α Equation (2)) and beta (β Equation (2)) to subtract the cross-talk signal from the E_mTotal_. The details on the development of the qFRET method can be found in a previous study on the development of the qFRET signal by Song et al. (2011) [25]. The method has been applied previously to determine kinetic values of protein–protein interactions, such as dissociation constant K_D_ and enzymatic constants k_cat_/K_M_, and applied to assess SUMO modification of viral proteins [25,52,59]. These discoveries could provide new insights for human-SARS-CoV-2 interactions, potentially a novel strategy for inhibiting host factors as anti-viral therapeutics development.

As the SARS and SARS-CoV-2 N proteins play critical roles in viral genome packaging and virion formation, our discovery that two critical activities of the N protein, oligomerization, and nucleus translocation, may indicate that the human PTM may play critical roles for the N protein in viral replication and viral RNA genome packaging. This requires future investigation. The blockage of host–virus interactions has become a promising approach for developing anti-viral therapeutics. This approach could have a very broad anti-viral spectrum for many viruses and low drug-resistance development potential as the human DNA polymerases have much higher fidelity than the viral reverse transcriptase or RNA polymerases [39]. This anti-viral strategy could be a significant supplement to the current mRNA vaccine and inhibitors for viral proteins.

## Figures and Tables

**Figure 1 viruses-15-01600-f001:**
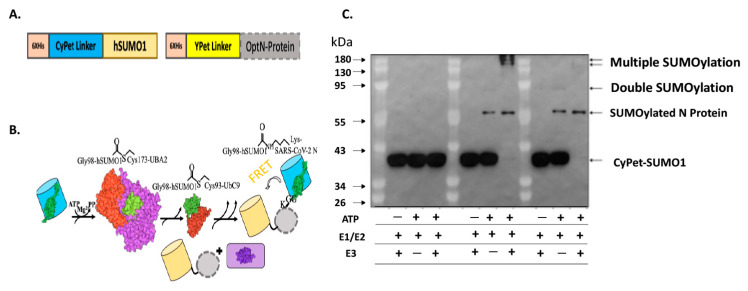
SUMOylation of SARS-CoV-2 N protein. (**A**) Diagram of the fluorescence fusion protein CyPet-SUMO1 and YPet-N for the qFRET-based SUMOylation assay. (**B**) In vitro SUMOylation assay with the FRET as a reporter signal. The fusion protein CyPet-SUMO is first bound to E1 activating enzyme, for the intermediate E1-Cypet-SUMO1 thioester bond at Cys173 to Gly98 on SUMO1. The SUMO is then transferred to the catalytic Cys-93 of E2 conjugating enzyme. The E3 ligase and target protein are said to non-covalently interact with the E2-SUMO1 complex. The CyPet-SUMO1 is then shuttled to a lysine on the target protein, to be covalently bound by an isopeptide bond. (**C**) The Western-blot of SUMOylated N protein in the in vitro reaction containing E1, E2 and E3 using a monoclonal anti-SUMO1 antibody.

**Figure 2 viruses-15-01600-f002:**
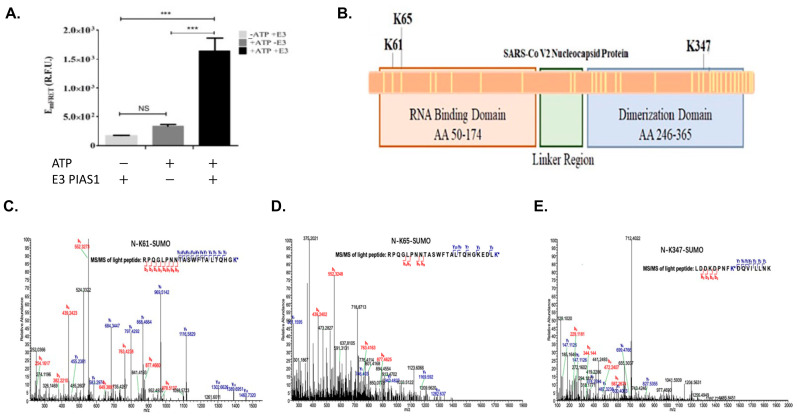
In vitro SUMOylation of SARS-CoV-2 N protein and SUMOylation site identification using Mass Spectrometry (MS). (**A**) The in vitro MS sample was measured for qFRET signal before processing for MS, with and without ATP or E3 PIAS1. (**B**) The illustration of the location of the three discovered lysines and a total of 31 lysines on the protein shown as yellow lines. (**C**) The spectrum was generated by Thermofisher Proteome DiscovererTM MS spectrum of peptide containing modified K61. (**D**) The spectrum of K65 peptide with trypsin cut on SUMO1, GG. (**E**) The spectrum of K347 was found in the same peptide, with SUMO1 peptide GG cut with trypsin. *p* values are *p* < 0.0001 *** and no significant difference (NS), *n* = 3.

**Figure 3 viruses-15-01600-f003:**
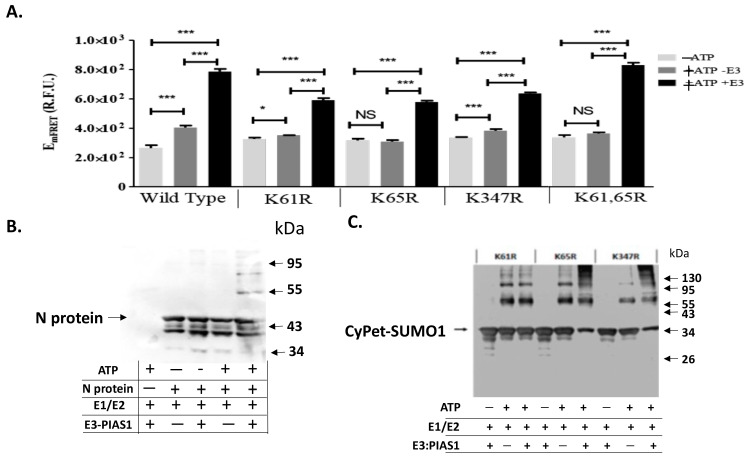
qFRET assay and Western-blot of in vitro SUMOylation of N protein and its Lys mutants. (**A**). SUMOyaltion assay of wildtype and K mutants of N protein using qFRET assay. Comparison of no ATP (−ATP), with no E3 ligase PIAS1 (−E3), and a complete reaction with ATP and E3 ligase PIAS1 (+ATP+E3). The reactions were performed under the same conditions, and the measurements were taken on the same instrument, Molecular Devices SpectraMax3^TM^. One-way ANOVA was performed on the data sets of −ATP/−E3/+ATP+E3, the −ATP was the control group. Tukey test was used as the post hoc analysis. *p* values are *p* < 0.0001 ***, *p* < 0.05 *, and no significant difference (NS), *n* = 3. (**B**) In vitro SUMOyaltion assay of N protein was probed with anti-N protein antibody in the Western-blot analysis. (**C**) The Western blot of various K-to-R mutants of N protein was probed with the anti-SUMO1 antibody.

**Figure 4 viruses-15-01600-f004:**
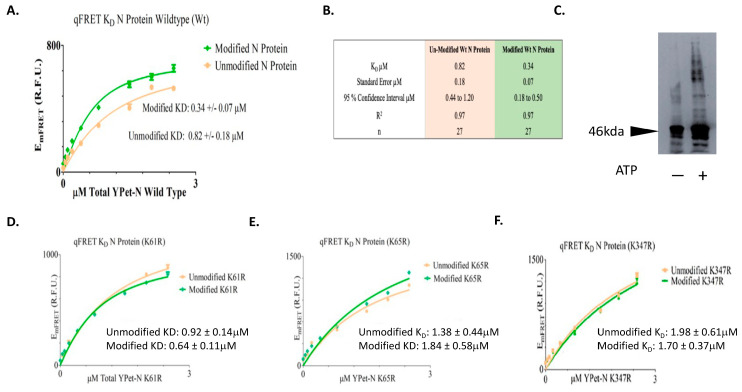
The K_D_ determinations of SUMOylated or un-SUMOylated N protein—N protein interactions using qFRET assay and aggregation assay of N protein with or without SUMO modification. (**A**) The binding curve of wildtype of N protein with or without SUMOylation. The Em_FRET_ signal fit of SUMO modified (Diamond/Green), and unmodified (Circle/Orange). (**B**) The summary of binding affinity of Un-SUMOylated and SUMOylated N protein to itself. (**C**) The aggregation assay of SARS-CoV-2 N protein with or without the SUMOs modification. The 10 μg of SARS-CoV-2 N protein was SUMOylated with SUMO1, Aos1/Uba2, Ubc9, and PIAS1 in the presence or absence of ATP, followed with disuccinimidyl suberate and Immunoblot with anti-N protein antibody. (**D**) The Binding curves of the SUMO-modified N protein mutants of (K61R). (**E**) The binding curves of the SUMO-modified N protein mutants of (K65R). (**F**) The binding curves of the SUMO-modified N protein mutants of (K347R). All plots and the fits were generated on GraphpadPrism5^TM^.

**Figure 5 viruses-15-01600-f005:**
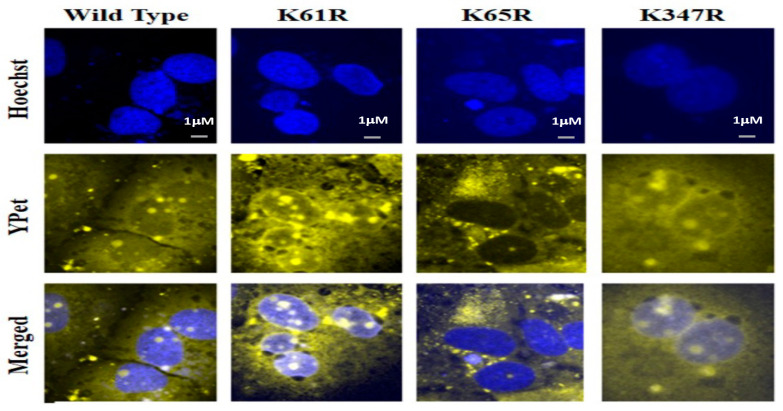
Subcellular localization determination of wildtype N protein and Lys mutants using a fluorescent microscope. The nuclear stain Hoechst was determined at 488 nm. The images of YPet-tagged-N proteins and their mutants were taken at 533 nm using the fluorescent microscope, Olympus BX43. Images were processed on ImageJ^TM^.

**Table 1 viruses-15-01600-t001:** The K_D_ values of the single mutant SARS-CoV-2 N protein with and without SUMOylation modification.

N Protein	K61R Unmodified	K61R Modified	K65R Unmodified	K65R Modified	K347R Unmodified	K347R Modified
K_D_ (µM)	0.92	0.64	1.38	1.84	1.98	1.70
Standard Error (µM)	0.14	0.11	0.44	0.58	0.61	0.37
95 % Confidence Interval (µM)	0.62 to 1.21	0.41 to 0.87	0.46 to 2.30	0.64 to 3.03	0.71 to 3.21	0.99 to 3.97
R^2^	0.99	0.98	0.95	0.91	0.97	0.97
n	27	27	27	27	27	27

**Table 2 viruses-15-01600-t002:** Primers listed for constructing N protein mutants in *E. coli*.

pET28B Primers	
K61Rfor	ccagcatggcagagaagacctgaaattt
K61Rrev	caggtcttctctgccatgctgggtcag
K65Rfor	gaagacctgagatttccgcgcggccag
K65Rrev	ctggccgcgcggaaatctcaggtcttc
K347Rfor	gatccgaattttcgagatcaggtgatt
K347Rrev	aatcacctgatctcgaaaattcggatc

**Table 3 viruses-15-01600-t003:** List of primers for mutations on N protein in HUH7 cells.

pCDNA3.1 Primers	
pcD_Ncwt_61For	ctcactcaacatggcagggaagacctt
pcD_Ncwt_61Rev	aaggtcttccctgccatgttgagtgag
pcD_Ncwt_65For	gaagaccttagattccctcgaggacaa
pcD_Ncwt_65Rev	ttgtcctcgagggaatctaaggtcttc
pcD_Ncwt_347For	aaagatccaaatttcagagatcaagtcatt
pcD_Ncwt_347Rev	aatgacttgatctctgaaatttggatcttt

## Data Availability

All the data can be available upon request to J.L. at Jiayu.liao@ucr.edu.

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
