# Peer review of "Human Post-Translational SUMOylation Modification of SARS-CoV-2 Nucleocapsid Protein Enhances Its Interaction Affinity with Itself and Plays a Critical Role in Its Nuclear Translocation"

_viruses, 2023, doi:10.3390/v15071600_

Round 1
Reviewer 1 Report
- Major comments:
In this study, Vipul Madahar et al. identified the SUMOylation sites of the SARS-CoV-2 N protein. They found that SUMO modification in N protein could increase the interaction affinity with itself, as well as showed that K65 residue was critical for N protein translocation to the nucleus by using qFRET/MS coupled method as well as immunofluorescence imaging.
The study is helpful for a better understanding of the human Post-translational Modification SUMOylation of SARS-CoV-2 Nucleocapsid protein in viral replication and pathogenesis.
- General concept comments
Here are some major considerations/suggestions for the study:
a. The rationale of the K-to-R mutations in the sites of 61, 65, and 347 of N protein. For example, why not mutate lysine to acidic amino acids or aromatic amino acids? K and R both belong to basic amino acids, and they are similar. K-to-R mutations may not be a good choice.
b. The data in Figures 4F and 4E is not consistent with the data in Table 1.
c. According to Figure 4E, 4F, and Table 1, it seems like the K65R mutation decreases the binding affinity of the N protein to itself, while the K61R mutation could increase the binding affinity of the N protein to itself?
d. What’s the rationale for doing the Lys mutant of 355 in the experiments of KD Determination and Cellular Translocation of N protein? Besides, the primers for K355R mutation are not included in Tables 2 and 3.
e. According to Figure 1C, it seems like the SUMOylation of N protein is also possible without E3 ligase? If so, how do you explain the absence of SUMOylation of N protein in columns 2 and 4 in Figure 3B?
f. When compared to Figures 3B and 3C, it seems like the three mutants actually enhanced the SUMOyaltion of N protein?
g. The description in Lines 433-436 is not consistent with Figure 4B.
h. The caption in Figure 4B is not consistent with the image of it.
i. K355R result is absent in Figure 5, according to lines 473 and 312. Throughout the study, where is the result of K355R?
j. Figure 4C, is the aggregation of SARS-CoV-2 N protein mediated by non-covalent interactions? Is the oligomerized SARS-CoV-2 N protein homogenous?
- Specific comments:
a. Lines 137-138, What do FLDD and FLAA in equation 1 stand for? Free donor and acceptor emissions, respectively?
b. Lines 155-156, samples were digested at a 1:100 ratio for the sample-to-enzyme ratio?
c. Line 281, here it should be Equation 2.
d. Line 364, typo of “s”?
e. Line 383, Figure 2B, the illustration of the location of the four discovered lysines? Well, only three were shown in Figure 2B.
f. Please provide high-resolution images of Figure 2C.
g. Label missing in Figures 4A and 4D.
h. Line 483, here, it should be K347R.
i. Is it possible that the granules observed in Figure 5 are in a position above the nucleus (on the top of the nucleus) rather than inside the nucleus?
j. Line 513, it should be Lys65 here?
k. Line 570, it should be Equation 1 here rather than Table 1?
l. Line 574, it should be Equation 2 here rather than Equation 3?

Extensive editing of English language required.
Author Response
Thank Reviewer 1 very much for such detailed comments on our manuscript! I have carefully read the comments and revised the manuscript as the following response,
Reviewer 1
- Major comments:
In this study, Vipul Madahar et al. identified the SUMOylation sites of the SARS-CoV-2 N protein. They found that SUMO modification in N protein could increase the interaction affinity with itself, as well as showed that K65 residue was critical for N protein translocation to the nucleus by using qFRET/MS coupled method as well as immunofluorescence imaging.
The study is helpful for a better understanding of the human Post-translational Modification SUMOylation of SARS-CoV-2 Nucleocapsid protein in viral replication and pathogenesis.
Thanks Reviewer 1 for the very positive comments!
- General concept comments
Here are some major considerations/suggestions for the study:
- The rationale of the K-to-R mutations in the sites of 61, 65, and 347 of N protein. For example, why not mutate lysine to acidic amino acids or aromatic amino acids? K and R both belong to basic amino acids, and they are similar. K-to-R mutations may not be a good choice.
The rational behind the K-to-R mutation was that this mutation is a conservative change that can preserve the N protein native structure as much as possible, except that R cannot be modified by SUMO peptide. Other mutations, such as K-to-D or K-to-F, may cause significant structural change of N protein itself and later the activity changes may be due to structural change.
- The data in Figures 4F and 4E is not consistent with the data in Table 1.
Thanks for pointing out this! The Fig4E&F have been revised.
- According to Figure 4E, 4F, and Table 1, it seems like the K65R mutation decreases the binding affinity of the N protein to itself, while the K61R mutation could increase the binding affinity of the N protein to itself?
Yes.
- What’s the rationale for doing the Lys mutant of 355 in the experiments of KD Determination and Cellular Translocation of N protein? Besides, the primers for K355R mutation are not included in Tables 2 and 3.
The Lys355 was initially used as a control. It’s SUMOylation was ambiguous, so we decided to drop the data for Lys355 in both KD and nuclear translocation assays.
- According to Figure 1C, it seems like the SUMOylation of N protein is also possible without E3 ligase? If so, how do you explain the absence of SUMOylation of N protein in columns 2 and 4 in Figure 3B?
Thanks for pointing this out! We checked the original record and found mistakes in labeling the figure legend. There was no ATP and no E3 PIAS1 in column 2. We have corrected it. In general, the SUMOylation of substrate could take place without E3 in vitro as there are high concentrations of E1, E2, and substrate. If the concentrations of E1, E2, and substrate are low, the SUMOylation reaction can be facilitated by the E3.
- When compared to Figures 3B and 3C, it seems like the three mutants actually enhanced the SUMOyaltion of N protein?
Good catch up! But the enzyme and substrate protein concentrations in these two experiments were different. The proteins in Figure 3B were much less than those in Figure 3C as to show the role of E3 and alternative SUMOylaiton sites, respectively.
- The description in Lines 433-436 is not consistent with Figure 4B.
Thanks for pointing out this labeling mistake! The label in Figure 4B was reversed. It has been corrected.
- The caption in Figure 4B is not consistent with the image of it.
Thanks! The caption in Figure 4B has been corrected.
- K355R result is absent in Figure 5, according to lines 473 and 312. Throughout the study, where is the result of K355R?
As explained before the K355 SUMOylation was ambiguous, so we dropped all the data for K355. We have revised the text contents in lines 473 and 312.
- Figure 4C, is the aggregation of SARS-CoV-2 N protein mediated by non-covalent interactions? Is the oligomerized SARS-CoV-2 N protein homogenous?
The aggregation of SARS-CoV-2 N tends to be stronger after SUMOyaltion. But to show the stronger aggregation of N protein in Figure 4C, the aggregated proteins were chemically cross-linked first to stabilize the aggregates and then run the gel. Yes, the oligomerized SARS-CoV-2 N protein was homogenous, with no precipitates.
- Specific comments:
- Lines 137-138, What do FLDDand FLAA in equation 1 stand for? Free donor and acceptor emissions, respectively?
FLDD stands for fluorescence emission of the donor when excited at donor excitation wavelength, and FLAA stands for fluorescence emission of the acceptor when excited at the acceptor excitation wavelength. The annotation has been added to the text.
- Lines 155-156, samples were digested at a 1:100 ratio for the sample-to-enzyme ratio?
To be more clear, it has been changed as “at a 1:100 ratio for the substrate-to-enzyme ratio”
- Line 281, here it should be Equation 2.
In line 281, it should still be Equation 1. But the next equation has been changed to Equation 2.
- Line 364, typo of “s”?
Thanks! It has been changed to “lysine residues”.
- Line 383, Figure 2B, the illustration of the location of the four discovered lysines? Well, only three were shown in Figure 2B.
Thanks! It was changed to “three”.
- Please provide high-resolution images of Figure 2C.
Yes. The high-resolution images of Figure 2C and Figure 2 is provided.
- Label missing in Figures 4A and 4D.
Thanks! They are added.
- Line 483, here, it should be K347R.
We actually examined all three Lys residues in nuclear translocation, other than just K347R. Therefore, we write each Lys site.
- Is it possible that the granules observed in Figure 5 are in a position above the nucleus (on the top of the nucleus) rather than inside the nucleus?
This could be a possibility. However, in the N K347R mutant protein image, the nuclear areas were very clear, suggesting that this possibility may not be true. It will be nice to have a image from a 3-D fluorescence microscope in the future.
- Line 513, it should be Lys65 here?
This is a general discussion without mentioning specific Lys in the sentence.
- Line 570, it should be Equation 1 here rather than Table 1?
Thanks for pointing out this! It was corrected.
- Line 574, it should be Equation 2 here rather than Equation 3?
Thanks for pointing out this! It was corrected.
Reviewer 2 Report
The study conducted by Vipul et al. focused on investigating the role of SUMOylation, a post-translational modification, in the function of the SARS-CoV-2 N protein. The N protein is known to play a crucial role in viral replication and the production of new virus particles. The authors employed a combination of qFRET/MS analysis and immunofluorescence imaging techniques to identify SUMOylation sites and explore the impact of SUMO modification on the N protein.
The findings of the study revealed that SUMOylation of the SARS-CoV-2 N protein significantly enhanced its interaction affinity with itself. This suggests that SUMOylation may facilitate the formation of N protein oligomers, which could be important for its function in viral replication. Additionally, the study identified a specific lysine residue, K65, that was found to be critical for the translocation of the N protein to the nucleus. This highlights the importance of the host human SUMOylation pathway in regulating the proper functioning of the N protein during viral replication and pathology. They suggest that targeting essential host pathways, including SUMOylation, could be a promising strategy for the development of antiviral therapeutics not only against SARS-CoV-2 but also other viruses. By interfering with the SUMOylation process and disrupting the proper functioning of viral proteins, it may be possible to hinder viral replication and reduce viral pathogenicity.
The findings presented in the manuscript are intriguing; however, there are several concerns and areas that require further clarification and experimental validation to strengthen the conclusions drawn.
Specific concerns:
- Figure 1B: The author should provide clarification in the figure legend regarding the schematic representation of CyPet-SUMO1 conjugated with YPet-N protein.
- Figure 1C: The significant decrease in the observed expression of SUMOylated proteins despite all SUMO1 proteins being switched to the conjugated SUMOylation form raises questions. The discrepancy should be addressed and explained by the author. Additionally, the detection of SUMOylated N protein using anti-SUMO1 antibody should be accompanied by an anti-N protein blot for clarity. The organization of the figure could be improved to clearly indicate which protein was used in each column of the in vitro SUMOylation assay.
- Line 348-350: The statement "The observed EmFRET signal for the SUMOylation of SARS-CoV-2 N protein showed some difference but not significant in the magnitude of EmFRET without or with the addition of ATP" contradicts the data presented in Figure 2A, where the SUMOylation level significantly increased in the presence of ATP (lane 1 vs. lane 3). The author should clarify whether the E2 enzyme was added in addition to the E3 PIAS1 in the reaction.
- Figure 3A: The data suggest that the K61R, K65R, and K347R mutants have a minor effect on N protein SUMOylation. However, it would be more appropriate to compare the mutants to the wild-type N protein rather than solely comparing them amongst themselves. The lack of a wild-type control in Figure 3C raises concerns about attributing specific functions to the mutants without appropriate comparison. Including a wild-type experiment in Figure 3C is crucial to provide a proper control and accurately interpret the results. Furthermore, the inclusion of the E1 and E2 enzymes in the in vitro SUMOylation assay should be specified.
- Figure 4D-4F: Similarly, the mutants show only minor effects on N protein SUMOylation. It is recommended to include the triple mutant (K61/K65/K347) in addition to the K61/K65 double mutant for a comprehensive analysis.
- Figure 5: The scale bar is missing, which should be included for proper interpretation of the results.
- The functional significance of N protein SUMOylation during SARS-CoV-2 replication needs to be addressed. If SUMOylation enhances the interaction affinity and alters translocation, the implications of N protein SUMOylation in viral replication should be discussed. This could involve exploring its role in viral assembly, RNA packaging, or interactions with host factors, which would contribute to a comprehensive understanding of SARS-CoV-2 replication.
Addressing these concerns and providing further experimental validation will strengthen the conclusions and provide a solid foundation for the implications of N protein SUMOylation in SARS-CoV-2 replication.
N/A
Author Response
Thank Reviewer 2 very much for such detailed comments on our manuscript! I have carefully read the comments and revised the manuscript as the following response,
Review 2
The study conducted by Vipul et al. focused on investigating the role of SUMOylation, a post-translational modification, in the function of the SARS-CoV-2 N protein. The N protein is known to play a crucial role in viral replication and the production of new virus particles. The authors employed a combination of qFRET/MS analysis and immunofluorescence imaging techniques to identify SUMOylation sites and explore the impact of SUMO modification on the N protein.
The findings of the study revealed that SUMOylation of the SARS-CoV-2 N protein significantly enhanced its interaction affinity with itself. This suggests that SUMOylation may facilitate the formation of N protein oligomers, which could be important for its function in viral replication. Additionally, the study identified a specific lysine residue, K65, that was found to be critical for the translocation of the N protein to the nucleus. This highlights the importance of the host human SUMOylation pathway in regulating the proper functioning of the N protein during viral replication and pathology. They suggest that targeting essential host pathways, including SUMOylation, could be a promising strategy for the development of antiviral therapeutics not only against SARS-CoV-2 but also other viruses. By interfering with the SUMOylation process and disrupting the proper functioning of viral proteins, it may be possible to hinder viral replication and reduce viral pathogenicity.
The findings presented in the manuscript are intriguing; however, there are several concerns and areas that require further clarification and experimental validation to strengthen the conclusions drawn.
Thanks, Reviewer 2, for the very detailed and encouraging comments!
Specific concerns:
- Figure 1B: The author should provide clarification in the figure legend regarding the schematic representation of CyPet-SUMO1 conjugated with YPet-N protein.
Yes. A more detailed figure legend has been added.
- Figure 1C: The significant decrease in the observed expression of SUMOylated proteins despite all SUMO1 proteins being switched to the conjugated SUMOylation form raises questions. The discrepancy should be addressed and explained by the author. Additionally, the detection of SUMOylated N protein using anti-SUMO1 antibody should be accompanied by an anti-N protein blot for clarity. The organization of the figure could be improved to clearly indicate which protein was used in each column of the in vitro SUMOylation assay.
Yes. The complete conjugation of CyPet-SUMO1 to YPet-N was explained in the text. We used both Anti-SUMO1 antibody (Figure 1C and Figure 3C) anti-N antibody (Figure 3B) in our Western blots. The more detailed reaction components were added to the figure legend.
- Line 348-350: The statement "The observed EmFRET signal for the SUMOylation of SARS-CoV-2 N protein showed some difference but not significant in the magnitude of EmFRET without or with the addition of ATP" contradicts the data presented in Figure 2A, where the SUMOylation level significantly increased in the presence of ATP (lane 1 vs. lane 3). The author should clarify whether the E2 enzyme was added in addition to the E3 PIAS1 in the reaction.
Thanks for pointing out this confusion! I have changed the sentence as “The observed EmFRET signal for the SUMOylation of SARS-CoV-2 N protein showed some difference but not significant in the magnitude of EmFRET without E3, but significant increase in the presence of E3”
- Figure 3A: The data suggest that the K61R, K65R, and K347R mutants have a minor effect on N protein SUMOylation. However, it would be more appropriate to compare the mutants to the wild-type N protein rather than solely comparing them amongst themselves. The lack of a wild-type control in Figure 3C raises concerns about attributing specific functions to the mutants without appropriate comparison. Including a wild-type experiment in Figure 3C is crucial to provide a proper control and accurately interpret the results. Furthermore, the inclusion of the E1 and E2 enzymes in the in vitro SUMOylation assay should be specified.
Thanks for the great suggestion! However, the main message that the Figure 3C intends to deliver is that even with each Lys mutation, the mutant N protein could still be SUMOylated, suggesting other SUMOyaltion sites. Therefore, including the wildtype of N protein here is not that critical, and in addition, the SUMOyaltion of N protein is shown in Figure 3B from Wester-blot and FRET assay in Figure 3A too.
- Figure 4D-4F: Similarly, the mutants show only minor effects on N protein SUMOylation. It is recommended to include the triple mutant (K61/K65/K347) in addition to the K61/K65 double mutant for a comprehensive analysis.
Thanks for the valuable suggestion! Actually, we thought about this, but consider that too many Lys mutations may change the N protein structure as SUMOylation substrate and it may not be easy to interpret the negative result. In addition, the given revision timeline is very short (10 days) and we could not conduct this experiment in such short time. We would certainly test this suggestion in the future.
- Figure 5: The scale bar is missing, which should be included for proper interpretation of the results.
Yes. The scale bar was added.
- The functional significance of N protein SUMOylation during SARS-CoV-2 replication needs to be addressed. If SUMOylation enhances the interaction affinity and alters translocation, the implications of N protein SUMOylation in viral replication should be discussed. This could involve exploring its role in viral assembly, RNA packaging, or interactions with host factors, which would contribute to a comprehensive understanding of SARS-CoV-2 replication.
Thanks for the valuable suggestions! I have added a paragraph to indicate the potential of our discovery in both basic research on viral replication and potential novel therapeutics development with significant advantages by blocking host-viral interactions at the end of the Discussion.
Addressing these concerns and providing further experimental validation will strengthen the conclusions and provide a solid foundation for the implications of N protein SUMOylation in SARS-CoV-2 replication.
Thanks Reviewer, for the very valuable comments and detailed suggestions! This is just the beginning of understanding the role of SUMOylation on SARS-CoV-2 N protein. We are certainly excited to explore more of SUMOylation on the SARS-CoV-2 replication in more detail in the future.
Reviewer 3 Report
The authors used FRET microscopy imaging and mass spectrometry analysis to identify the SUMOylation sites in SARS-CoV-2 nucleocapsid (N) protein and tested its biological function in transfected, uninfected cells. They found 3 SUMOylation in N protein: K61, K65, and K347. qFRET analysis showed that mutation of all SUMOylation sites reduced self-interaction of N protein while microscopy analysis revealed the importance of K65 in the nuclear localization of N protein. In general, the FRET and MS analysis were well done but the microscope images in Fig 5 could be improved. I wonder if some quantification could be performed to clearly show that K65 SUMOylation is required for the nuclear localization of N protein. Are these SUMOylation sites are conserved in other coronavirus N proteins, and could they also be SUMOylated? If so, SUMOylation of N protein in coronavirus infected cells could also be tested in a BSL2 lab. In lines 513-14, it is stated that K347 SUMOylation may be critical for N protein dimerization in vivo. However, this statement is not supported by the results in Figure 4. I also strongly suggest that the manuscript is thoroughly checked for English with a native speaker. There are typos and grammatical errors in several places.
I also strongly suggest that the manuscript is thoroughly checked for English with a native speaker. There are typos and grammatical errors in several places.
Author Response
Thank Reviewer 3 very much for such detailed comments on our manuscript! I have carefully read the comments and revised the manuscript as the following response,
Review 3
The authors used FRET microscopy imaging and mass spectrometry analysis to identify the SUMOylation sites in SARS-CoV-2 nucleocapsid (N) protein and tested its biological function in transfected, uninfected cells. They found 3 SUMOylation in N protein: K61, K65, and K347. qFRET analysis showed that mutation of all SUMOylation sites reduced self-interaction of N protein while microscopy analysis revealed the importance of K65 in the nuclear localization of N protein. In general, the FRET and MS analysis were well done but the microscope images in Fig 5 could be improved. I wonder if some quantification could be performed to clearly show that K65 SUMOylation is required for the nuclear localization of N protein. Are these SUMOylation sites are conserved in other coronavirus N proteins, and could they also be SUMOylated? If so, SUMOylation of N protein in coronavirus infected cells could also be tested in a BSL2 lab. In lines 513-14, it is stated that K347 SUMOylation may be critical for N protein dimerization in vivo. However, this statement is not supported by the results in Figure 4
Thanks reviewer for the positive comments!
We have chosen one typical nucleus from each image and determined the fluorescence intensities using Image J software as follows,
|
Proteins |
Nucleus fluorescence intensity |
|
N WT |
73.184 |
|
K347R |
85.976 |
|
K65R |
28.124 |
|
K61R |
100.671 |
This result indicates a significant reduction of N K65R protein in nucleus.
In order to determine the SUMOylated Lye residue is conserved among the coronaviruses, we have retrived the N proteins for other coronal viruses, such as SARS, human Coronavirus 229E, human Coronavirus HKU1, human Coronavirus OC43, human Coronavirus NL63, and MERS, and did alignment. We found that in the region of 61-65 amino acid, all viruses have at least one or two Lys residues; and in the region of 347 amino acid,
all viruses also have at least one or two Lys residues, suggesting the conservation of SUMOylation sites in these two regions. The potential conserved mechanism of SUMOylation for coronaviruses is worth further investigation.
Because the interaction affinity increases were insignificant after K347 SUMOylation, I deleted the sentence in lines 513-14 mentioning K347 SUMOyaltion as critical for N protein dimerization in vivo. There may be other SUMOylation sites in this region, such as K355, for dimerization.
Comments on the Quality of English Language
I also strongly suggest that the manuscript is thoroughly checked for English with a native speaker. There are typos and grammatical errors in several places.
Thanks for the reviewer’s suggestion! We did ask a native English speaker to revise the manuscript.
Round 2
Reviewer 1 Report
I think that the manuscript has been improved, and the authors have addressed most of my concerns except that all figures should be updated to high-resolution ones.

Fine.
Author Response
Reviewer 1
I think that the manuscript has been improved, and the authors have addressed most of my concerns except that all figures should be updated to high-resolution ones.
Thanks the Reviewer 1 for the positive feedback! I have changed the figures to high-resolution(600 dpi) ones. Please let me know if they are fine or not.
Reviewer 3 Report
The authors have adequately addressed my concerns.
Please correct line 79-86.
Author Response
Reviewer 3
Please correct line 79-86.
Thanks for catching this! I have revised these sentences.